# The Relation between Milk Lactose Concentration and the Rumination, Feeding, and Locomotion Behavior of Early-Lactation Dairy Cows

**DOI:** 10.3390/ani14060836

**Published:** 2024-03-08

**Authors:** Ramūnas Antanaitis, Karina Džermeikaitė, Justina Krištolaitytė, Akvilė Girdauskaitė, Samanta Arlauskaitė, Kotryna Tolkačiovaitė, Walter Baumgartner

**Affiliations:** 1Large Animal Clinic, Veterinary Academy, Lithuanian University of Health Sciences, Tilžės Str. 18, LT-47181 Kaunas, Lithuania; karina.dzermeikaite@lsmu.lt (K.D.); justina.kristolaityte@stud.lsmu.lt (J.K.); akvile.girdauskaite@lsmu.lt (A.G.); samanta.arlauskaite@lsmu.lt (S.A.); kotryna.tolkaciovaite@stud.lsmu.lt (K.T.); 2University Clinic for Ruminants, University of Veterinary Medicine, Veterinaerplatz 1, A-1210 Vienna, Austria; walter.baumgartner@vetmeduni.ac.at

**Keywords:** milk lactose, precision dairy farming, innovation, behavior

## Abstract

**Simple Summary:**

Precision livestock farming involves the use of real-time monitoring technologies to manage the smallest unit of production, focusing specifically on individual animals via sensor technology. This study hypothesizes that higher in-line milk lactose concentrations are indicative of enhanced dairy cow behaviors—including increased rumination, feeding, and locomotion activities—reflecting superior overall health and well-being. It posits that fluctuations in milk lactose levels have a substantial impact on the physiological and behavioral responses of dairy cows, thereby affecting their milk yield and composition. The objective was to explore this potential relationship, examining how in-line milk lactose concentrations might influence these specific behavioral patterns in dairy cows. We observed significant differences among groups regarding milk yields, milk protein concentrations, other chews, rumination chews, boluses, and changes in activity levels. Specifically, cows with a milk lactose concentration of ≥4.70% showed a 16.14% increase in milk yields, other chews, rumination chews, and an increase in boluses. However, these cows also experienced a decrease in milk protein concentrations and in activity levels.

**Abstract:**

This study hypothesizes that higher in-line milk lactose concentrations are indicative of enhanced dairy cow behaviors—including increased rumination, feeding, and locomotion activities—reflecting superior overall health and well-being. It posits that fluctuations in milk lactose levels have a substantial impact on the physiological and behavioral responses of dairy cows, thereby affecting their milk yields and compositions. Each cow’s milk lactose, fat, protein, and fat-to-protein ratio were continuously monitored using the BROLIS HerdLine in-line milk analyzer (Brolis Sensor Technology, Vilnius, Lithuania). The RumiWatch noseband sensor (RWS; ITIN + HOCH GmbH, Fütterungstechnik, Liestal, Switzerland) was employed to measure the biomarkers of the rumination, feeding, and locomotion behavior. The measurements were recorded over 5 days at the same time (during morning milking). A total of 502 cows were examined. During these 5 days, 2510 measurements were taken. Based on the lactose content in their milk, the cows were divided into two categories: the first group consisted of cows with milk lactose levels below 4.70%, while the second group included cows with milk lactose levels of 4.70% or higher. Our study showed that cows with higher milk lactose concentrations (≥4.70%) produced significantly more milk (16.14% increase) but had a lower milk protein concentration (5.05% decrease) compared to cows with lower lactose levels. These cows also exhibited changes in rumination and feeding behaviors, as recorded by the RWS: there was an increase in the mastication and rumination behaviors, evidenced by a 14.09% rise in other chews and a 13.84% increase in rumination chews, along with a 16.70% boost in bolus activity. However, there was a notable 16.18% reduction in their physical activity, as measured by the change in time spent walking.

## 1. Introduction

Precision livestock farming (PLF) refers to the application of real-time monitoring technologies for the management of the most granular production unit, essentially targeting individual animals through sensor technology. PLF offers significant opportunities for value creation across various stakeholders, primarily serving as an effective management resource for farmers. It enhances the capacity to boost animal welfare, efficiency, and health, while also reducing environmental impacts [1]. In PLF, notable real-time monitoring systems include the RumiWatch system (RWS) and the BROLIS HerdLine in-line milk analyzer. The RumiWatch system (ITIN + HOCH GmbH, Fütterungstechnik, Liestal, Switzerland) is an innovative device that integrates a noseband sensor with a pedometer to create a highly functional system known for its utility, sensitivity, and specificity [2]. The noseband sensor of the RWS was specifically developed and proven effective as a tool for automatically recognizing rumination and feeding activities in dairy cows housed in stables [2]. Büchel [3] found that the RWS accurately captures the behavior of individual animals over an extended period. Despite its frequent application in various research projects, comprehensive validations of the RWS, particularly for barn-fed cows, have been primarily documented by Zehner et al. [2] and Ruuska et al. [4]. The BROLIS HerdLine in-line milk analyzer (Brolis Sensor Technology, Vilnius, Lithuania) is an innovative sensor, which is able to identify lactose, as well as fat–protein ratios in a cow’s milk. During each milking cycle, the analyzer continuously measures the composition of each cow’s milk. This ‘mini-spectroscope’ can be mounted on the milking stalls or on a milking robot in the milk line and requires no additional reagents or maintenance [5]. Milk serves as an ideal medium for assessing the health of dairy cows due to its non-invasive collection method and ease of access, making it a common tool for detecting ketosis and other issues related to production [6]. Milk parameters originate from components in both blood and feed, and comprehending the relationships among these elements in feed, blood, and milk is crucial for assessing the health and production status of animals [7].

Milk lactose, a significant component of bovine milk solids, is influenced considerably by the health of the udder as well as the cow’s metabolic and energy balance [8]. Contemporary dairy farming often aims for high milk yields, leading to health issues in cows [9]. Lactose plays a crucial role as an osmotic agent in milk, significantly influencing the transfer of water from the bloodstream into the milk. Lactose production occurs within the udder, where blood glucose is absorbed through the basal membrane of the mammary epithelial cells and then converted into lactose [10]. During lactation, approximately 20% of a dairy cow’s circulating blood glucose is transformed into lactose [11]. Consequently, a reduced lactose level leads to a decreased production in milk volume [12]. In the age of big data, advanced milking technologies now offer daily insights into milk production from individual cows [8]. Studies have shown that milk lactose levels tend to decrease, while somatic cell counts (SCCs) increase in cases of clinical and subclinical mastitis [13]. Therefore, tracking lactose levels in milk can serve as a diagnostic tool for mastitis, evident from a notable reduction during inflammation [14]. From our past findings, it can be concluded that the in-line measurement of milk lactose concentrations can serve as an indicator of the health status of dairy cows. Cows with a higher lactose concentration (≥4.70%) were found to be more active by 54.47% and had a lower risk of mastitis, as evidenced by a lower electrical conductivity of the milk and somatic cell counts, as well as fewer metabolic disorders, as determined by the milk fat-to-protein ratio (F/P). Low lactose levels may signal the presence of mastitis (milk SCC ≥ 100,000/mL) and metabolic disorders (such as subclinical ketosis and subclinical acidosis), as indicated by the milk F/P ratio [15]. Regularly monitoring these lactose concentrations aids in farm-quality control and management, facilitating the detection of low-quality milk and energy inefficiencies [16].

Therefore, further exploration into the use of early milk sampling and the potential integration of novel data sources is warranted to enhance the accuracy of early warning systems. Additionally, to ensure the robustness of predictive models against variations in farm management and feed compositions, it is crucial to gather and evaluate more data from diverse farms before implementing such models in practice [5]. Understanding the linkage between lactose concentrations and cow behaviors can help farms to pre-emptively address issues like mastitis, which is known to alter lactose levels, before they escalate into more severe health problems. The early detection of such conditions via lactose monitoring can facilitate prompt treatment, reducing the risk of spreading diseases and improving milk quality. The use of PLF tools for monitoring lactose levels thus not only aids in enhancing animal welfare but also supports the sustainability of dairy operations by optimizing productivity and ensuring the production of high-quality milk. This approach exemplifies how integrating detailed behavioral and physiological data can revolutionize farm-management strategies, leading to more efficient and humane dairy farming practices. 

The hypothesis of the current study was that there is a relationship between in-line milk lactose concentrations and the rumination, feeding, and locomotion behavior of dairy cows. The aim of this study was to investigate the relationship between in-line milk lactose concentrations and behaviors such as rumination, feeding, and locomotion, as well as variables including milk yields, milk fat, milk proteins, milk fat–protein ratios, other chews, rumination chews, eating chews, drinking gulps, boluses, chews per minute, activities, and activity changes of dairy cows.

## 2. Materials and Methods

### 2.1. Animal Housing Conditions of This Study

Throughout this study, we adhered to the Lithuanian Law on Animal Welfare and Protection, with this study receiving approval under the number PK012858. This research took place in Lithuania (coordinates: 55.819156, 23.773541) from 1 July to 31 July 2023. The dairy cows were housed in free-stall barns equipped with ventilation systems and were provided with a total mixed ration (TMR) tailored to their physiological requirements year-round. Feeding times were set at 06:00 and 18:00 daily, offering a TMR designed for high-producing, multiparous cows. The diet mainly included 25% of corn silage, 5% of alfalfa grass hay, 20% of grass silage, 15% of sugar beet pulp silage, 30% of grain concentrate mash, and 5% of a mineral mix, formulated to satisfy or surpass the nutritional demands of a 500 kg Holstein cow yielding 37 kg of milk per day. The ration’s chemical profile was specified as follows: dry matter (DM) at 48.80%, neutral detergent fiber at 28.20% of DM, acid detergent fiber at 19.80% of DM, non-fiber carbohydrates at 38.70% of DM, crude protein at 15.80% of DM, and a net lactation energy of 1.60 Mcal/kg. Milking occurred twice daily, at 05:00 and 17:00, via a parlor system. Out of 1160 clinically examined cows, 502 were chosen for this study, specifically those in their second or later lactation periods and within the first 5 to 30 days post-calving. The average weight of these cows was 550 kg ± 45 kg, and the average energy-corrected milk yield (with 4.2% fat and 3.6% protein) per cow per lactation was 12,500 kg.

### 2.2. Registration of Parameters

During this study, we recorded the milk composition using the BROLIS HerdLine in-line milk analyzer (Brolis Sensor Technology, Vilnius, Lithuania) and monitored the rumination, feeding, and locomotion behavior with the help of the RumiWatch noseband sensor (RWS; ITIN + HOCH GmbH, Fütterungstechnik, Liestal, Switzerland).

#### 2.2.1. Registration of Milk Composition

Each cow’s milk lactose, fat, protein, and fat-to-protein ratio were continuously monitored using the BROLIS HerdLine in-line milk analyzer. This analyzer features a distinctive gallium antimonide (GaSb) widely tunable external-cavity laser-based in-line spectrometer operating within the 2100–2400 nm wavelength range. Milk flow throughout the milking process was consistently tracked in transmission mode, allowing for the collection of molecular absorption spectra. These spectra were analyzed to ascertain the levels of the primary milk constituents. The analyzer performed continuous composition measurements of the milk from each cow during every milking session. Installed directly on the milking parlor stalls or within the milk line of a milking robot, this ‘mini-spectroscope’ operates without the need for any extra reagents or maintenance. For each milking session, the analyzer continuously records the composition of each cow’s milk at 5-s intervals. The concentrations of fat, protein, and lactose are then weighted according to the milk flow, averaging these dynamics to produce single values that represent the entirety of the milking process. The accuracy of the milk analyzer was assessed at a Eurofins laboratory, yielding values of the root mean square error of prediction of 0.21% for fat, 0.19% for protein, and 0.19% for lactose. 

#### 2.2.2. Registration of Rumination, Feeding, and Locomotion Behavior

The RumiWatch noseband sensor (RWS) was employed to measure the biomarkers of the rumination, feeding, and locomotion behavior. The RWS consists of a fluid-filled pressure tube and a noseband halter equipped with an integrated pressure sensor. This sensor sends pressure signals to a data recorder, which is attached to the halter and is housed within a durable plastic enclosure. Additionally, the device features a slot for a memory card and an acceleration sensor capable of monitoring three-dimensional head movements. Both the acceleration and pressure data are logged as binary files at a 10 Hz frequency. The RumiWatch system includes a wireless data transmitter connected to the halter, facilitating the real-time collection of data. The RWS software’s (RumiWatch Manager 2 software (V. 2.2.0.0)) core algorithms are responsible for accurately categorizing the behavioral data from the 10 Hz pressure readings across various time intervals (Table 1). 

#### 2.2.3. Duration of Parameter Registration

The RWS was implemented from 1 June to 31 July 2023. The initial fortnight, spanning 1 June to 14 July 2023, was designated as an adjustment phase for the cows to become familiar with the RWS. The actual monitoring using RWS started on 14 June 2023 and continued until 31 July 2023. Data recording took place hourly, every day. Throughout the period from 1 June to 14 June 2023, during each milking session, the BROLIS HerdLine in-line milk analyzer continuously recorded the composition of each cow’s milk from start to finish. 

### 2.3. Groups Creation

The measurements were recorded over 5 days at the same time (during morning milking). A total of 502 cows were examined. During these 5 days, 2510 measurements were taken. According to the literature [17] on lactose threshold, we created two groups: the first group consisted of cows with milk lactose levels below 4.70%, while the second group included cows with milk lactose levels of 4.70% or higher.

### 2.4. Statistical Analysis

This study’s statistical analysis was conducted with IBM SPSS Statistics for Windows, Version 25.0, developed by IBM Corp. in 2017, located in Armonk, New York, NY, USA. To verify the normal distribution of the data, the Shapiro–Wilk test was employed. The findings are presented as the mean and standard error of the mean (SEM), with a significance threshold set at 0.05 (*p* < 0.05) for assessing probability. Furthermore, to explore statistical associations among the variables under study, a Pearson correlation analysis was performed. A linear regression equation was used to ascertain the statistical relationship between the in-line milk lactose and other parameters. This relationship was deemed statistically significant (*p* = 0.05) if the probability value fell below 0.05. 

## 3. Results

### 3.1. Descriptive Statistics

According to our results, we found significant differences between groups in milk yields, milk proteins, other chews, rumination chews, boluses, and activity changes. Additionally, no differences were found between the groups in terms of milk fat, milk fat-to-protein ratios, eating chews 1, eating chews 2, drinking gulps, chews per minute, and activities (Table 2).

Milk Yield (MY): In the group of cows with a higher milk lactose concentration, we observed a significantly higher milk production (*p* < 0.001). The average MY in Group I (ML < 4.70%) was 12.94 (±3.75) kg/session, while in Group II (ML ≥ 4.70%), it was 15.43 (±3.85) kg/session. In cows with an ML of ≥4.70%, the MY was 16.14% higher compared to the group with an ML of less than 4.70%.

Milk protein (MP): We found a significant (*p* < 0.001) decrease of 5.05% in the milk protein concentration among cows with higher milk lactose concentrations. The average milk protein (MP) concentration in Group I (ML < 4.70%) was 3.33% (±0.04), while in Group II (ML ≥ 4.70%), it was 3.17% (±0.20).

Other Chews (OCs): We discovered that the parameter of OCs was significantly higher (*p* < 0.01), by 14.09%, in Group II (ML ≥ 4.70%) compared to Group I (ML < 4.70%). The average value of OCs in Group I was 163.24 (n/h) (±3.46), while in Group II, it was 190.02 (n/h) (±8.08).

Rumination Chews (RCs): Our study found that the parameter of RCs was significantly higher (*p* < 0.01), by 13.84%, in Group II (ML ≥ 4.70%) compared to Group I (ML < 4.70%). The average value of RCs in Group II was 1280.80 (n/h) (±52.28), while in Group I, it was 1103.58 (n/h) (±23.37).

Bolus (B): Our findings indicate a significant (*p* < 0.001) increase, of 16.70%, in boluses in Group II (ML ≥ 4.70%) compared to Group I (ML < 4.70%). The average value of boluses in Group II was 22.69 (n/h) (±0.92), whereas in Group I, it was 18.90 (n/h) (±0.38).

Activity Changes (ACs): Our findings show a significant (*p* < 0.001) decrease, of 16.18%, in activity changes in Group II (ML ≥ 4.70%) compared to Group I (ML < 4.70%). The average value of ACs in Group II was 19.53 (min/h) (±0.23), whereas in Group I, it was 23.32 (min/h) (±0.08).

### 3.2. Correlation between Investigated Parameters

Correlations between milk lactose and the other investigated parameters are presented in Table 3. We observed a weak significant positive correlation between milk lactose concentrations and milk yields (r = 0.366, *p* < 0.001). Higher lactose concentrations in milk are associated with an increased milk production (Figure 1). Also, we found a weak but significant correlation between milk lactose and milk proteins (r = −0.210, *p* < 0.01); milk fat/protein ratios (F/P) (r = −0.086, *p* < 0.01); eating chews 1 (r = 0.049, *p* < 0.05); rumination chews (r = 0.045, *p* < 0.05); boluses (r = 0.065, *p* < 0.001); chews per minute (r = 0.084, *p* < 0.001); and activity changes (r = 0.113, *p* < 0.001) (Figure 2).

## 4. Discussion

The real-time tracking of milk lactose levels multiple times a day allows for the observation of its fluctuations across different physiological states and throughout the duration of cow diseases. The widespread implementation of precision farming technologies facilitates the daily documentation of individual milk profiles and variations in specific milk components. This can aid in the early detection of health issues and the initiation of prompt treatments [18].

This study aimed to investigate the relationship between in-line milk lactose concentrations and the rumination, eating, and locomotion behavior of dairy cows. Our findings indicate significant differences between groups in terms of milk yields, milk protein concentrations, other chews, rumination chews, boluses, and activity changes. According to our results, a positive correlation was observed between the milk lactose concentration and milk yield. Cows with a milk lactose concentration of ≥4.70% exhibited a 16.14% increase in milk production. The correlations observed between lactose yields and milk yields align with the findings of Miglior et al. [19] and Sneddon et al. [20]. Our findings are in agreement with the 0.40 estimate from New Zealand data presented by Sneddon et al. [20]. Milk lactose, a significant component of bovine milk solids, is influenced considerably by the health of the udder, as well as the cow’s metabolic and energy balance. Given its connections to various biological and physiological factors, the literature offers insights into milk lactose, focusing on its chemical characteristics, inheritability, and genetic links to health traits [21]. Furthermore, lactose has been identified as a marker for both subclinical and clinical ketosis [22], and it offers the most accurate assessment of energy balance [12]. Consequently, the lactose concentration in milk can serve as a marker for mastitis, with notable reductions observed during inflammation [13]. The regular monitoring of lactose levels can aid in farm-quality control and management, assisting in detecting low-quality milk [23] and pinpointing energy inefficiencies [14]. The measurement of lactose levels is now commonly employed as a criterion for the early detection and management of herds [24]. Lactose plays a crucial role as an osmotic agent in milk, significantly influencing the transfer of water into milk from the bloodstream. Therefore, reduced lactose levels lead to a decrease in the overall volume of milk produced [12]. Glucose stimulates the growth of cells and the production of lactose in the mammary epithelial cells of dairy cows. Protein kinase B alpha functions as a metabolic regulator in the mammary gland of dairy cows, facilitating the impact of glucose on lactose production [25]. Currently, it is widely agreed that lactose primarily functions as an osmolyte in milk, drawing more water into the milk as lactose synthesis increases. Consequently, an increase in lactose synthesis results in a higher volume of milk production. This mechanism does not alter the total amount of other milk components, such as proteins and solids, leaving them unchanged. As a result, while the overall milk yield increases, the concentration of its constituents decreases [20]. Moreover, our findings indicate that cows with a milk lactose concentration exceeding 4.70% exhibited a 5.05% decrease in their milk protein concentration, underscoring the slight yet statistically significant inverse relationship between milk lactose and milk protein concentrations (r = −0.210, *p* < 0.01). According to the literature, genetic correlations between the lactose percentage (LP) and the fat and protein percentages across days in milk (DIM) for first-lactation cows were identified, which paralleled the lactation curve for milk yields with an early peak in lactation [26]. Haile-Mariam and Pryce [27] observed, in 2017, a change in the genetic correlation between the LP and protein percentage, shifting from moderately positive (0.30) in early lactation to moderately negative (−0.24) in late lactation. Also, more significant correlations in third-lactation cows than in first-lactation cows were discovered [27].

Blood glucose levels and energy balance in cows have a positive association with lactose production, particularly in high-yielding breeds, as noted by Reist et al. [9] and Larsen and Moyes [28]. Lemosquet et al. [29] highlighted that the availability of post-hepatic blood glucose might indirectly regulate milk production, positioning blood glucose as a direct influencer of lactose yields. Therefore, it is crucial to recognize the reliance of milk production on milk lactose, emphasizing that glucose absorption from the blood for lactose synthesis is a metabolic imperative in dairy breeds. Indeed, in high-yielding cows, udder needs are managed through homeorhesis, ensuring a stable milk composition, even when body reserves are utilized [30].

In our previous study, we discovered that cows with higher milk lactose concentrations were significantly more active (increasing by 54.47%) and had a lower risk of developing subclinical acidosis (SARA), as demonstrated by a 2.52% increase in the reticulo-ruminal pH and a 9.86% extension in the rumination time [15]. Antanaitis et al. [5] found that cows suffering from acidosis were less active compared to healthy ones. The definition of SARA based on rumen pH is subject to debate [31], but previous research [32,33] has shown that a rumen pH below 6 is indicative of SARA. Supporting the findings of Antanaitis et al. [34], it has been documented that SARA-affected cows experience shorter rumination periods.

Evaluating the correlation between milk lactose concentrations, cow energy balance, and its effectiveness in predicting cow reproductive success, our findings align with those reported by Buckley et al. [35]. They found that milk lactose levels, reflecting blood glucose levels, serve as a reliable indicator for assessing the energy balance in cows.

Reksen et al. [36] discovered that cows in their second-lactation period with elevated milk lactose levels during the initial 8 weeks post-calving exhibited an earlier luteal response. When examining the link between milk lactose concentrations and cow energy balance, alongside its utility in evaluating reproductive success in cows, our findings align with those presented by Buckley et al. [35]. They indicated that the milk lactose content, reflecting blood glucose levels, serves as a marker not only for assessing cows’ energy balance but also for their reproductive performance [15].

We found a 16.18% decrease in activity changes (changes in the total amount of time spent walking in a specific recording period, expressed as minutes) in cows with a milk lactose concentration above 4.70%. In our previous research, we observed that cows exhibiting higher concentrations of milk lactose were 54.47% more active [15]. In the current study, we did not identify significant differences between the groups. Coulon et al. [37] conducted research to assess the impact of walking activities on milk production and energy status in dairy farms, utilizing tie-stall housing for the cows. It has been observed that dairy cows exhibit behavioral signs of illnesses during mastitis, with alterations in the activity, lying time, and feeding behavior being the primary focus of scientific study [38]. Such behavioral changes are believed to be triggered by pain or other adverse experiences [39]. In the literature, we did not find any information about the impact of milk lactose on changes in cow activities.

This study is subject to several limitations. First, the small sample size may limit the generalizability of our findings to a broader population. Future research would benefit from including a larger and more diverse cohort of cows to validate these results. Second, data collection was confined to the summer months. Seasonal variations significantly impact the physiological conditions of dairy cows, which could in turn influence milk compositions and cow behaviors. Consequently, the outcomes observed in this study might not be representative of other seasons. It is recommended that subsequent studies expand the data collection period to include different seasons, ensuring a more comprehensive understanding of the dynamics explored.

## 5. Conclusions

Our study highlights significant associations between behavioral patterns and physiological changes in dairy cows, suggesting that monitoring behavioral changes could directly indicate health issues like mastitis. While our initial hypothesis explored the utility of lactose content as an indirect marker, feedback and advancements in precision livestock farming technologies suggest a more direct approach might be equally or more effective. Thus, we recommend focusing on direct behavioral observations facilitated by current PLF technologies as primary indicators of physiological health. This shift acknowledges the direct link between observed behavioral changes and underlying health conditions, potentially streamlining early detection and management strategies for conditions such as mastitis.

From a practical viewpoint, we recommend regularly monitoring milk lactose levels as indicators of behavioral and physiological changes. Adjust diets for cows with lactose levels of ≥4.70% to maintain milk protein quality while supporting increased milk production. Implement measures to encourage physical activities in cows showing decreased movements, potentially improving their overall well-being and performance.

## Figures and Tables

**Figure 1 animals-14-00836-f001:**
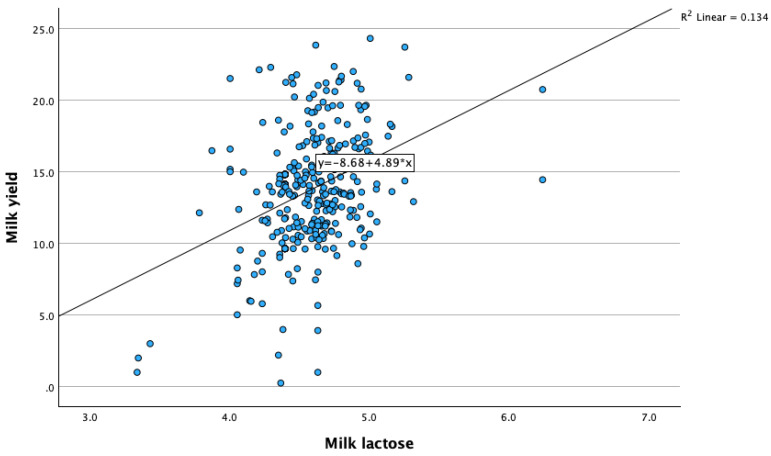
Relationship between milk lactose concentrations and milk yields.

**Figure 2 animals-14-00836-f002:**
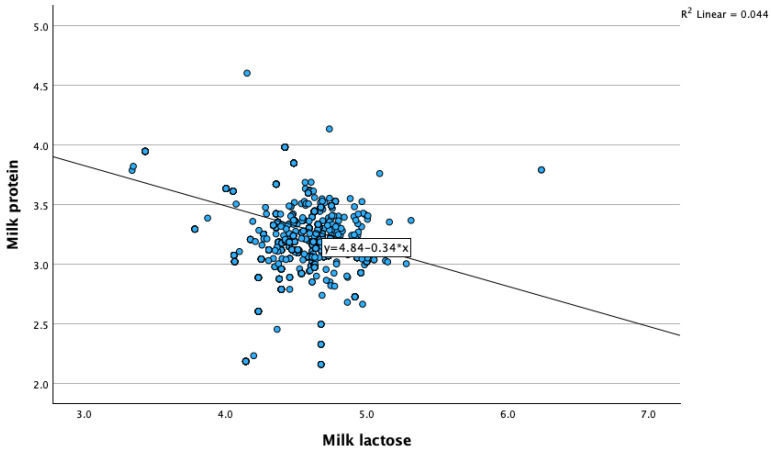
Relationship between milk lactose concentrations and milk proteins.

**Table 1 animals-14-00836-t001:** Description of investigated parameters.

Indicator	Description	Registration Source
Milk yield (MY) (kg/session)	Milk yield per milking session (kg)	DeLaval milking system
Milk lactose (ML) (%)	Milk lactose concentration (%)	BROLIS HerdLine in-line milk analyzer
Milk fat (MF) (%)	Milk fat concentration (%)	BROLIS HerdLine in-line milk analyzer
Milk protein (MP) (%)	Milk protein concentration (%)	BROLIS HerdLine in-line milk analyzer
Milk fat-to-protein ratio (F/P)	Fat–protein ratio in the milk	BROLIS HerdLine in-line milk analyzer
Rumination chews (RCs) (n/h)	Chewing using the mouth during rumination to mechanically break up regurgitated material into smaller pieces	RumiWatch sensor
Eating chews 1 (EC1) (n/h)	Number of chews performed while the head is positioned downward during the chosen summary interval	RumiWatch sensor
Eating chews 2 (EC2) (n/h)	Number of chews performed with head positioned upward during the chosen summary interval	RumiWatch sensor
Drinking gulps (DGs) (n/h)	The cumulative number of swallows during the drinking process	RumiWatch sensor
Bolus (B) (n/h)	The overall quantity of sips or swallows made during the act of drinking	RumiWatch sensor
Chews per minute (CPM) (n/min)	Chewing movements occurring during rumination following regurgitation per minute	RumiWatch sensor
Activity (min/h)	The total amount of time spent walking in a specific recording period expressed as minutes	RumiWatch sensor
Activity change (AC) (min/h)	Change in the total amount of time spent walking in a specific recording period expressed as minutes	RumiWatch sensor
Other chews (OCs) (n/h)	Total amount of mastication chews and fear bites during eating	RumiWatch sensor

**Table 2 animals-14-00836-t002:** Descriptive statistics of investigated parameters.

		N	Mean	Std. Deviation	Std. Error	95% Confidence Interval for Mean		Minimum	Maximum	*p* Value
						Lower Bound	Upper Bound			
Milk yield (MY) kg/session	Group I	2101	12.94	3.75	0.22	12.51	13.38	0.25	23.85	<0.001
	Group II	409	15.43	3.52	0.34	14.76	16.11	8.59	24.31	
	Total	2510	13.61	3.85	0.19	13.23	14.00	0.25	24.31	
Milk lactose (L) %	Group I	2101	4.46	0.18	0.01	4.45	4.47	3.33	4.69	<0.001
	Group II	409	4.86	0.17	0.01	4.85	4.88	4.69	6.23	
	Total	2510	4.52	0.23	0.01	4.51	4.53	3.33	6.23	
Milk fat (F) %	Group I	2101	4.47	0.85	0.01	4.43	4.51	2.19	8.70	0.439
	Group II	409	4.43	0.86	0.04	4.35	4.52	2.58	6.34	
	Total	2510	4.46	0.86	0.01	4.43	4.50	2.19	8.70	
Milk protein (P) %	Group I	2101	3.33	0.04	0.01	3.32	3.35	2.16	4.60	<0.001
	Group II	409	3.17	0.20	0.01	3.15	3.19	2.66	4.13	
	Total	2510	3.31	0.38	0.01	3.29	3.32	2.16	4.60	
Milk fat-to- protein ratio (F/P)	Group I	2101	1.36	0.39	0.01	1.34	1.38	0.67	4.02	0.096
	Group II	409	1.39	0.24	0.01	1.37	1.42	0.82	1.86	
	Total	2510	1.37	0.37	0.01	1.35	1.38	0.67	4.02	
Other chews (OCs) (n/h)	Group I	2101	163.24	157.08	3.46	156.45	170.03	0	1484	0.003
	Group II	409	190.02	152.96	8.08	174.12	205.92	0	1047	
	Total	2510	167.21	156.73	3.18	160.95	173.46	0	1484	
Rumination chews (RCs) (n/h)	Group I	2101	1103.58	1060.21	23.37	1057.74	1149.42	0	4090	0.003
	Group II	409	1280.80	989.18	52.28	1177.99	1383.62	0	3991	
	Total	2510	1129.85	1051.67	21.40	1087.89	1171.82	0	4090	
Eating chews 1 (EC1) (n/h)	Group I	2101	373.25	668.44	14.73	344.35	402.16	0	4064	0.285
	Group II	409	414.39	690.90	36.51	342.57	486.20	0	3715	
	Total	2510	379.35	671.83	13.67	352.54	406.16	0	4064	
Eating chews 2 (EC2) (n/h)	Group I	2101	404.07	572.87	12.63	379.30	428.84	0	3634	0.072
	Group II	409	462.39	525.41	27.76	407.78	517.01	0	2620	
	Total	2510	412.72	566.36	11.52	390.12	435.32	0	3634	
Drinking gulps (DGs) (n/h)	Group I	2101	181.47	348.36	7.68	166.41	196.53	0	2593	0.464
	Group II	409	196.20	367.46	19.42	158.00	234.39	0	2528	
	Total	2510	183.65	351.22	7.14	169.64	197.67	0	2593	
Bolus (B) (n/h)	Group I	2101	18.90	17.60	0.38	18.14	19.66	0	88	<0.001
	Group II	409	22.69	17.46	0.92	20.88	24.50	0	86	
	Total	2510	19.46	17.63	0.35	18.76	20.16	0	88	
Chews per minute (CPM) (n/min)	Group I	2101	4.85	11.53	0.25	4.35	5.35	0	233	0.257
	Group II	409	5.59	10.79	0.57	4.47	6.71	0	71	
	Total	2510	4.96	11.43	0.23	4.50	5.42	0	233	
Activity (min/h)	Group I	2101	64.04	53.44	1.17	61.73	66.35	0	408	0.851
	Group II	409	63.49	37.49	1.98	59.59	67.38	8	192	
	Total	2510	63.96	51.38	1.04	61.91	66.01	0	408	
Activity changes (ACs) (min/h)	Group I	2101	20.53	3.83	0.08	20.37	20.70	7	31	<0.001
	Group II	409	19.53	4.50	0.23	19.07	20.00	11	31	
	Total	2510	20.39	3.95	0.08	20.23	20.54	7	31	

Group I—milk lactose concentration of <4.70%; Group II—milk lactose concentration of ≥4.70%; N—number of measurements; *p*—significance level.

**Table 3 animals-14-00836-t003:** Correlations between investigated parameters.

		MY	F	P	F/P	L	EC1	EC2	OCs	RCs	EC1	EC2	DGs	B	CPM	Activity	ACs
MY	Pearson Correlation	1	−0.181 **	0.207 **	−0.263 **	0.366 **	−0.008	−0.034	−0.058	0.255 **	−0.017	−0.034	−0.013	0.238 **	0.189 **	0.036	0.003
F	Pearson Correlation	−0.181 **	1	0.006	0.847 **	−0.175 **	−0.045 *	0.058 **	0.102 **	0.046 *	−0.040 *	0.060 **	−0.034	0.048 *	0.058 **	0.137 **	0.049 *
P	Pearson Correlation	0.207 **	0.006	1	−0.492 **	−0.210 **	0.040	0.107 **	0.004	0.210 **	0.063 **	0.156 **	0.116 **	0.176 **	0.211 **	0.273 **	0.053 **
F/P	Pearson Correlation	−0.263 **	0.847 **	−0.492 **	1	−0.086 **	−0.055 **	−0.009	0.101 **	−0.069 **	−0.061 **	−0.028	−0.075 **	−0.049 *	−0.058 **	−0.009	0.012
L	Pearson Correlation	0.366 **	−0.175 **	−0.210 **	−0.086 **	1	0.049 *	0.042 *	−0.007	0.045 *	0.028	0.018	0.019	0.065 **	0.084 **	−0.034	0.113 **
EC1	Pearson Correlation	−0.008	−0.045 *	0.040	−0.055 **	0.049 *	1	0.506 **	0.128 **	−0.326 **	0.948 **	0.544 **	0.927 **	−0.344 **	−0.198 **	0.568 **	0.538 **
EC2	Pearson Correlation	−0.034	0.058 **	0.107 **	−0.009	0.042 *	0.506 **	1	0.316 **	−0.316 **	0.480 **	0.978 **	0.465 **	−0.301 **	−0.142 **	0.734 **	0.693 **
OCs	Pearson Correlation	−0.058	0.102 **	0.004	0.101 **	−0.007	0.128 **	0.316 **	1	−0.250 **	0.118 **	0.259 **	0.101 **	−0.253 **	−0.030	0.560 **	0.435 **
	Sig. (2-tailed)	0.312	<0.001	0.836	<0.001	0.719	<0.001	<0.001		<0.001	<0.001	<0.001	<0.001	<0.001	0.135	<0.001	<0.001
RCs	Pearson Correlation	0.255 **	0.046 *	0.210 **	−0.069 **	0.045 *	−0.326 **	−0.316 **	−0.250 **	1	−0.305 **	−0.315 **	−0.288 **	0.915 **	0.729 **	−0.143 **	−0.094 **
EC1	Pearson Correlation	−0.017	−0.040 *	0.063 **	−0.061 **	0.028	0.948 **	0.480 **	0.118 **	−0.305 **	1	0.527 **	0.902 **	−0.325 **	−0.176 **	0.548 **	0.507 **
EC2	Pearson Correlation	−0.034	0.060 **	0.156 **	−0.028	0.018	0.544 **	0.978 **	0.259 **	−0.315 **	0.527 **	1	0.530 **	−0.312 **	−0.163 **	0.755 **	0.631 **
DGs	Pearson Correlation	−0.013	−0.034	0.116 **	−0.075 **	0.019	0.927 **	0.465 **	0.101 **	−0.288 **	0.902 **	0.530 **	1	−0.309 **	−0.179 **	0.589 **	0.463 **
B	Pearson Correlation	0.238 **	0.048*	0.176 **	−0.049 *	0.065 **	−0.344 **	−0.301 **	−0.253 **	0.915 **	−0.325 **	−0.312 **	−0.309 **	1	0.746 **	−0.158 **	−0.048 *
CPM	Pearson Correlation	0.189 **	0.058 **	0.211 **	−0.058 **	0.084 **	−0.198 **	−0.142 **	−0.030	0.729 **	−0.176 **	−0.163 **	−0.179 **	0.746 **	1	−0.010	0.169 **
Activity	Pearson Correlation	0.036	0.137 **	0.273 **	−0.009	−0.034	0.568 **	0.734 **	0.560 **	−0.143 **	0.548 **	0.755 **	0.589 **	−0.158 **	−0.010	1	0.653 **
ACs	Pearson Correlation	0.003	0.049 *	0.053 **	0.012	0.113 **	0.538 **	0.693 **	0.435 **	−0.094 **	0.507 **	0.631 **	0.463 **	−0.048 *	0.169 **	0.653 **	1

MY—milk yield; F—milk fat; P—milk protein; F/P—milk fat-to-protein ratio; L—milk lactose; EC1—eating chews 1; EC2—eating chews 2; DGs—drinking gulps; B—bolus; CPM—chews per minute; ACs—activity changes. * Correlation is significant at the 0.05 level (2-tailed). ** Correlation is significant at the 0.01 level (2-tailed).

## Data Availability

The data provided in this study can be found in the publication.

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
