# Peer review of "The Relation between Milk Lactose Concentration and the Rumination, Feeding, and Locomotion Behavior of Early-Lactation Dairy Cows"

_animals, 2024, doi:10.3390/ani14060836_

Round 1

Reviewer 1 Report

Comments and Suggestions for Authors

General comments:

 The manuscript animals-2891648, entitled "The Relation Between In-line Milk Lactose Concentration And Rumination, Eating and Locomotion Behavior Of Dairy Cows" with Mr. Antanaitis as first author deals with an interesting topic within the emerging field of precision livestock farming in dairy cattle. The manuscript is relatively structured, in most cases carefully formulated and presents interesting ideas. However, some sections need to be revised and major mistakes must be corrected especially in the results and discussion section. Various comments for changes to the text could be find in the following section.

Detailed comments:

Line 2              The word “And” should not be written in capital letters

Line 3            The word “Of” should not be written in capital letters

Line 8              The email address of S.A. is not entered correctly

Line 11            Why did you underline the full stop?

Line 13            Maybe you can insert a colon after the term “simple summary” like after “Abstract”?

Line 15            What is the basis for this hypothesis? Please describe the possible causal relationship in simple words.

Line 24            What is the basis for this hypothesis? Please describe the possible causal relationship in simple words.

Line 35            Why did you change the font size within the Abstract?

Line 42            Please add the term “behavior” in the key word list

Line 45            General comment: The introduction should be rewritten, as it is very unstructured and does not lead the reader to the topic as intended. Please pay attention to a logical sequence in the introduction! I would suggest starting with a short section on PLF (including RumiWatch). Then I would be useful to describe the milk ingredients (including Milk lactose). Finally, you must make a connection between the two topics and describe the research gap.

Line 53                        This section is very redundant. For example, you mention the non-invasive method three times in just a few sentences. Please shorten the section and avoid repetition.

Line 72            You cite reference 7 very often here. Are there any other references on methodology that you could cite?

Line 88            What is the meaning of the parenthesis (.)?

Line 103          Please harmonize within the manuscript: “in-line” or “inline”?

Line 111                      In the introduction, you established a link between lactose content and health. The correlation between health and behavior was only barely illuminated. Please add a few more sentences to support your hypothesis.

Line 128          Please harmonize within the manuscript one or two decimal numbers?

Line 140          Please explain the abbreviation “GaSb”.

Line 164          Please explain the abbreviation “RWC”.

Table 1           The table contains many empty spaces and could therefore be presented more compactly.

The definition of “other chews” should be placed at the end of the table. The term “other chews” is included twice.

The terms and definitions are still unclear to me. For example, how can the eating time be specified in the unit (n/h) and not in minutes? Please check the contents of the table again!

Line 168       The abbreviation RWS was introduce before.

Line 168        Why is the data collection in July 2023 described in another section? (line 120)

Line 180          Please explain in more detail, why the threshold of 4.70% is relevant for your examination.

Line 194          Why did you use a semicolon here?

Table 2         The table is too large. Maybe you can use landscape format?

                    Could you also provide the values for milk lactose?

                    Why are different terms used here than in the Materials and        Methods section (e.g. EC2)?

Line 198          In this section, not only the significant results should be emphasized. It is equally interesting to highlight the non-significant results. Please extend this section considerably.

Line 219          Why haven't you included a correlation table here where you can also read the non-significant values? Please also provide the other results and describe them in the text.

Line 228          You should mention in the text, that the r² is very low and could not explain very much

Line 230          General comment: There are missing some information on potential relationships between lactose content and behavior? Why should the behavior might be affected by different lactose levels? Please add some more details.

Line 230          The limitations of the study should be described in a short section (e.g. small sample size; data collection only in summer)

Line 232          The first five lines provide basic information and should therefore be included in the introduction.

Line 245          A correlation of 0.366 should not be described as strong. Please provide a source for the classification...

Line 247        Why are you citing a study here that compares fat and lactose content, while you are looking at milk yield and lactose content?

Line 264          Did you investigate the influence of the lactation number in your study?

Line 300          How would you explain this different results in your studies?

Line 303          What is the practical impact of your findings for dairy farmers? Which benefit do they receive, if they use your method? Please add some more information

Line 315          Why did you change the font size here?

Line 327          There is missing the name of the journal

Line 329          There is missing the journal number and the pages

Line 331          Please use the abbreviated journal name

Line 337          Why do you use the term “et al.”. Please cite all authors of the publication.

Line 343          Is it really the correct abbreviation? “of”?

Line 351          Please use the abbreviated journal name

Line 355          Is it really the correct abbreviation? “of”?

Line 360          Is it really the correct abbreviation? “of”?

Line 374          Please use the abbreviated journal name

Line 379          Please delete the asterisk after the title.

Line 382          Is it really the correct abbreviation? “and”?

Author Response

Reviewer: General comments: The manuscript animals-2891648, entitled "The Relation Between In-line Milk Lactose Concentration And Rumination, Eating and Locomotion Behavior Of Dairy Cows" with Mr. Antanaitis as first author deals with an interesting topic within the emerging field of precision livestock farming in dairy cattle. The manuscript is relatively structured, in most cases carefully formulated and presents interesting ideas. However, some sections need to be revised and major mistakes must be corrected especially in the results and discussion section. Various comments for changes to the text could be find in the following section.

Authors: Authors are very thankful for the comments, which help us to improve the manuscript. All changes proposed have been included in the manuscript and highlighted in yellow and track changes.  

Best Regards, 

Prof. Ramunas Antanaitis 

Detailed comments:

Reviewer: Line 2              The word “And” should not be written in capital letters 

Authors: Corrected

Reviewer: Line 3            The word “Of” should not be written in capital letters 

Authors: Corrected

Reviewer: Line 8              The email address of S.A. is not entered correctly

Authors: we corrected to – “[email protected] (S.A.)”

Reviewer: Line 11            Why did you underline the full stop?

Authors: corrected

Reviewer: Line 13            Maybe you can insert a colon after the term “simple summary” like after “Abstract”?

Authors: Corrected to – “Simple Summary:”

Reviewer:  Line 15            What is the basis for this hypothesis? Please describe the possible causal relationship in simple words.

Authors: We corrected to – “The hypothesis of the current study was that there is a relationship between in-line milk lactose concentration and the parameters of rumination, eating, and locomotion”

Reviewer: Line 24            What is the basis for this hypothesis? Please describe the possible causal relationship in simple words.

Authors: we corrected to – “The hypothesis of the current study was that there is a relationship between in-line milk lactose concentration and the rumination, eating, and locomotion behavior of dairy cows”

Reviewer: Line 35            Why did you change the font size within the Abstract?

Authors: Corrected

Reviewer: Line 42            Please add the term “behavior” in the key word list

Authors: added

Reviewer: Line 45            General comment: The introduction should be rewritten, as it is very unstructured and does not lead the reader to the topic as intended. Please pay attention to a logical sequence in the introduction! I would suggest starting with a short section on PLF (including RumiWatch). Then I would be useful to describe the milk ingredients (including Milk lactose). Finally, you must make a connection between the two topics and describe the research gap.

Authors: We revised the entire introduction section and organized it into the following parts: 1. A brief overview of Precision Livestock Farming (PLF), including the RumiWatch system and BROLIS Sensor. 2. Identification of milk composition, with a focus on milk lactose. 3. We established a link between these two topics.

Reviewer:  Line 53                        This section is very redundant. For example, you mention the non-invasive method three times in just a few sentences. Please shorten the section and avoid repetition.

Authors: Corrected, we removed repetition of information and made this section schorten.

Reviewer: Line 72            You cite reference 7 very often here. Are there any other references on methodology that you could cite?

Authors: We revised this text and included contributions from other authors – “The RumiWatch system (ITIN + HOCH GmbH, Fütterungstechnik, Liestal, Switzerland) is an innovative device that integrates a noseband sensor with a pedometer to create a highly functional system known for its utility, sensitivity, and specificity 1.   Berckmans, D. General Introduction to Precision Livestock Farming. Animal Frontiers 2017, 7, 6–11, doi:10.2527/af.2017.0102.

  1. Zehner, N.; Umstätter, C.; Niederhauser, J.J.; Schick, M. System Specification and Validation of a Noseband Pressure Sensor for Measurement of Ruminating and Eating Behavior in Stable-Fed Cows. Computers and Electronics in Agriculture 2017, 136, 31–41, doi:10.1016/j.compag.2017.02.021.
  2. Büchel, S.; Sundrum, A. Short Communication: Decrease in Rumination Time as an Indicator of the Onset of Calving. Journal of Dairy Science 2014, 97, 3120–3127, doi:10.3168/jds.2013-7613.
  3. Ruuska, S.; Kajava, S.; Mughal, M.; Zehner, N.; Mononen, J. Validation of a Pressure Sensor-Based System for Measuring Eating, Rumination and Drinking Behaviour of Dairy Cattle. Applied Animal Behaviour Science 2016, 174, 19–23, doi:10.1016/j.applanim.2015.11.005.
  4. Antanaitis, R.; Džermeikaitė, K.; Januškevičius, V.; Šimonytė, I.; Baumgartner, W. In-Line Registered Milk Fat-to-Protein Ratio for the Assessment of Metabolic Status in Dairy Cows. Animals 2023, 13, 3293, doi:10.3390/ani13203293.
  5. Pires, J.A.A.; Larsen, T.; Leroux, C. Milk Metabolites and Fatty Acids as Noninvasive Biomarkers of Metabolic Status and Energy Balance in Early-Lactation Cows. Journal of Dairy Science 2022, 105, 201–220, doi:10.3168/jds.2021-20465.
  6. Liu, P.; He, X.; Yang, X.L.; Hou, X.L.; Han, J.B.; Han, Y.H.; Nie, P.; Fang, H.; Du, X.H. Bioactivity Evaluation of Certain Hepatic Enzymes in Blood Plasma and Milk of Holstein Cows. Pak Vet J 2012.
  7. Costa, A.; Egger-Danner, C.; Mészáros, G.; Fuerst, C.; Penasa, M.; Sölkner, J.; Fuerst-Waltl, B. Genetic Associations of Lactose and Its Ratios to Other Milk Solids with Health Traits in Austrian Fleckvieh Cows. Journal of Dairy Science 2019, 102, 4238–4248, doi:10.3168/jds.2018-15883.
  8. Reist, M.; Erdin, D.; von Euw, D.; Tschuemperlin, K.; Leuenberger, H.; Chilliard, Y.; Hammon, H.M.; Morel, C.; Philipona, C.; Zbinden, Y.; et al. Estimation of Energy Balance at the Individual and Herd Level Using Blood and Milk Traits in High-Yielding Dairy Cows1,2. Journal of Dairy Science 2002, 85, 3314–3327, doi:10.3168/jds.S0022-0302(02)74420-2.
  9. Osorio, J.S.; Lohakare, J.; Bionaz, M. Biosynthesis of Milk Fat, Protein, and Lactose: Roles of Transcriptional and Posttranscriptional Regulation. Physiological Genomics 2016, 48, 231–256, doi:10.1152/physiolgenomics.00016.2015.
  10. Cant, J.P.; Trout, D.R.; Qiao, F.; Purdie, N.G. Milk Synthetic Response of the Bovine Mammary Gland to an Increase in the Local Concentration of Arterial Glucose. Journal of Dairy Science 2002, 85, 494–503, doi:10.3168/jds.S0022-0302(02)74100-3.
  11. Vilas Boas, D.F.; Vercesi Filho, A.E.; Pereira, M.A.; Roma Junior, L.C.; El Faro, L. Association between Electrical Conductivity and Milk Production Traits in Dairy Gyr Cows. Journal of Applied Animal Research 2017, 45, 227–233, doi:10.1080/09712119.2016.1150849.
  12. Pyörälä, S. Indicators of Inflammation in the Diagnosis of Mastitis. Veterinary Research 2003, 34, 565–578, doi:10.1051/vetres:2003026.
  13. Rigout, S.; Lemosquet, S.; van Eys, J.E.; Blum, J.W.; Rulquin, H. Duodenal Glucose Increases Glucose Fluxes and Lactose Synthesis in Grass Silage-Fed Dairy Cows. Journal of Dairy Science 2002, 85, 595–606, doi:10.3168/jds.S0022-0302(02)74113-1.
  14. Televičius, M.; Juozaitiene, V.; Malašauskienė, D.; Antanaitis, R.; Rutkauskas, A.; Urbutis, M.; Baumgartner, W. Inline Milk Lactose Concentration as Biomarker of the Health Status and Reproductive Success in Dairy Cows. Agriculture 2021, 11, 38, doi:10.3390/agriculture11010038.
  15. Qiao, F.; Trout, D.R.; Xiao, C.; Cant, J.P. Kinetics of Glucose Transport and Sequestration in Lactating Bovine Mammary Glands Measured in Vivo with a Paired Indicator/Nutrient Dilution Technique. Journal of Applied Physiology2005, 99, 799–806, doi:10.1152/japplphysiol.00386.2004.
  16. Antanaitis, R.; Juozaitienė, V.; Jonike, V.; Baumgartner, W.; Paulauskas, A. Milk Lactose as a Biomarker of Subclinical Mastitis in Dairy Cows. Animals 2021, 11, 1736, doi:10.3390/ani11061736.
  17. Antanaitis, R.; Juozaitienė, V.; Malašauskienė, D.; Televičius, M.; Urbutis, M.; Baumgartner, W. Influence of Calving Ease on In-Line Milk Lactose and Other Milk Components. Animals 2021, 11, 842, doi:10.3390/ani11030842.
  18. Miglior, F.; Sewalem, A.; Jamrozik, J.; Bohmanova, J.; Lefebvre, D.M.; Moore, R.K. Genetic Analysis of Milk Urea Nitrogen and Lactose and Their Relationships with Other Production Traits in Canadian Holstein Cattle. Journal of Dairy Science 2007, 90, 2468–2479, doi:10.3168/jds.2006-487.
  19. Sneddon, N.; Lopez-Villalobos, N.; Davis, S.; Hickson, R.; Shalloo, L. Genetic Parameters for Milk Components Including Lactose from Test Day Records in the New Zealand Dairy Herd. New Zealand Journal of Agricultural Research2015, 58, 97–107, doi:10.1080/00288233.2014.978482.
  20. Costa, A.; Lopez-Villalobos, N.; Sneddon, N.W.; Shalloo, L.; Franzoi, M.; De Marchi, M.; Penasa, M. Invited Review: Milk Lactose—Current Status and Future Challenges in Dairy Cattle. Journal of Dairy Science 2019, 102, 5883–5898, doi:10.3168/jds.2018-15955.
  21. Steen, A.; ØSterås, O.; Grønstøl, H. Evaluation of Additional Acetone and Urea Analyses, and of the Fat-Lactose-Quotient in Cow Milk Samples in the Herd Recording System in Norway. Journal of Veterinary Medicine Series A 1996,43, 181–191, doi:10.1111/j.1439-0442.1996.tb00443.x.
  22. Leitner, G.; Lavi, Y.; Merin, U.; Lemberskiy-Kuzin, L.; Katz, G. Online Evaluation of Milk Quality According to Coagulation Properties for Its Optimal Distribution for Industrial Applications. Journal of Dairy Science 2011, 94, 2923–2932, doi:10.3168/jds.2010-3946.
  23. Berry, D.P.; Lee, J.M.; Macdonald, K.A.; Stafford, K.; Matthews, L.; Roche, J.R. Associations Among Body Condition Score, Body Weight, Somatic Cell Count, and Clinical Mastitis in Seasonally Calving Dairy Cattle. Journal of Dairy Science 2007, 90, 637–648, doi:10.3168/jds.S0022-0302(07)71546-1.
  24. Lin, Y.; Sun, X.; Hou, X.; Qu, B.; Gao, X.; Li, Q. Effects of Glucose on Lactose Synthesis in Mammary Epithelial Cells from Dairy Cow. BMC Vet Res 2016, 12, 81, doi:10.1186/s12917-016-0704-x.
  25. Shahbazkia, H.R.; Aminlari, M.; Tavasoli, A.; Mohamadnia, A.R.; Cravador, A. Associations among Milk Production Traits and Glycosylated Haemoglobin in Dairy Cattle; Importance of Lactose Synthesis Potential. Vet Res Commun 2010, 34, 1–9, doi:10.1007/s11259-009-9324-2.
  26. Haile-Mariam, M.; Pryce, J.E. Genetic Parameters for Lactose and Its Correlation with Other Milk Production Traits and Fitness Traits in Pasture-Based Production Systems. Journal of Dairy Science 2017, 100, 3754–3766, doi:10.3168/jds.2016-11952.
  27. Larsen, M.L.V.; Wang, M.; Norton, T. Information Technologies for Welfare Monitoring in Pigs and Their Relation to Welfare Quality®. Sustainability 2021, 13, 692, doi:10.3390/su13020692.
  28. Lemosquet, S.; Delamaire, E.; Lapierre, H.; Blum, J.W.; Peyraud, J.L. Effects of Glucose, Propionic Acid, and Nonessential Amino Acids on Glucose Metabolism and Milk Yield in Holstein Dairy Cows. Journal of Dairy Science2009, 92, 3244–3257, doi:10.3168/jds.2008-1610.
  29. Zhao, S.; Min, L.; Zheng, N.; Wang, J. Effect of Heat Stress on Bacterial Composition and Metabolism in the Rumen of Lactating Dairy Cows. Animals 2019, 9, 925, doi:10.3390/ani9110925.
  30. Kleen, J.L.; Cannizzo, C. Incidence, Prevalence and Impact of SARA in Dairy Herds. Animal Feed Science and Technology 2012, 172, 4–8, doi:10.1016/j.anifeedsci.2011.12.003.
  31. Li, G.; Ma, X.; Deng, L.; Zhao, X.; Wei, Y.; Gao, Z.; Jia, J.; Xu, J.; Sun, C. Fresh Garlic Extract Enhances the Antimicrobial Activities of Antibiotics on Resistant Strains in Vitro. Jundishapur J Microbiol 2015, 8, e14814, doi:10.5812/jjm.14814.
  32. Dann, H.M.; Morin, D.E.; Bollero, G.A.; Murphy, M.R.; Drackley, J.K. Prepartum Intake, Postpartum Induction of Ketosis, and Periparturient Disorders Affect the Metabolic Status of Dairy Cows*. Journal of Dairy Science 2005, 88, 3249–3264, doi:10.3168/jds.S0022-0302(05)73008-3.
  33. Antanaitis, R.; Juozaitienė, V.; Malašauskienė, D.; Televičius, M. Can Rumination Time and Some Blood Biochemical Parameters Be Used as Biomarkers for the Diagnosis of Subclinical Acidosis and Subclinical Ketosis? Veterinary and Animal Science 2019, 8, 100077, doi:10.1016/j.vas.2019.100077.
  34. Buckley, F.; O’Sullivan, K.; Mee, J.F.; Evans, R.D.; Dillon, P. Relationships Among Milk Yield, Body Condition, Cow Weight, and Reproduction in Spring-Calved Holstein-Friesians. Journal of Dairy Science 2003, 86, 2308–2319, doi:10.3168/jds.S0022-0302(03)73823-5.
  35. Reksen, O.; Havrevoll, Ø.; Gröhn, Y.T.; Bolstad, T.; Waldmann, A.; Ropstad, E. Relationships Among Body Condition Score, Milk Constituents, and Postpartum Luteal Function in Norwegian Dairy Cows. Journal of Dairy Science2002, 85, 1406–1415, doi:10.3168/jds.S0022-0302(02)74208-2.
  36. Coulon, J.B.; Pradel, P.; Cochard, T.; Poutrel, B. Effect of Extreme Walking Conditions for Dairy Cows on Milk Yield, Chemical Composition, and Somatic Cell Count1. Journal of Dairy Science 1998, 81, 994–1003, doi:10.3168/jds.S0022-0302(98)75660-7.
  37. Medrano-Galarza, C.; Gibbons, J.; Wagner, S.; de Passillé, A.M.; Rushen, J. Behavioral Changes in Dairy Cows with Mastitis. Journal of Dairy Science 2012, 95, 6994–7002, doi:10.3168/jds.2011-5247.
  38. Chapinal, N.; de Passillé, A.M.; Rushen, J. Correlated Changes in Behavioral Indicators of Lameness in Dairy Cows Following Hoof Trimming. Journal of Dairy Science 2010, 93, 5758–5763, doi:10.3168/jds.2010-3426.

[2]. The noseband sensor of the RWS has been specifically developed and proven effective as a tool for automatically recognizing rumination and feeding activities in dairy cows housed in stables [2]. Büchel [3] found that the RWS accurately captures the behavior of individual animals over an extended period. Despite its frequent application in various research, comprehensive validations of the RWS, particularly for barn-fed cows, have been primarily documented by Zehner et al.[2] and Ruuska et al. [4].”

Reviewer: Line 88            What is the meaning of the parenthesis (.)?

Authors: Deleted

Reviewer: Line 103          Please harmonize within the manuscript: “in-line” or “inline”?

Authors- Corrected to “in-line” in whole manuscript.

Reviewer: Line 111                      In the introduction, you established a link between lactose content and health. The correlation between health and behavior was only barely illuminated. Please add a few more sentences to support your hypothesis.

Authors: We corrected to – “The aim of this study was to investigate the relationship between in-line milk lactose concentration and behaviors such as rumination, feeding, and locomotion, as well as variables including milk yield, milk fat, milk protein, milk fat-protein ratio, other chews, rumination chews, eating chews, drinking gulps, bolus, chews per minute, activity, and activity change of dairy cows”

Reviewer: Line 128          Please harmonize within the manuscript one or two decimal numbers?

Authors: Corrected

Reviewer: Line 140          Please explain the abbreviation “GaSb”.

Authors: Corrected to – “gallium antimonide (GaSb)”

Reviewer: Line 164          Please explain the abbreviation “RWC”.

Authors: We corrected to – “The RWS software's..”

Reviewer: Table 1           The table contains many empty spaces and could therefore be presented more compactly.

Authors: Corrected and deleted empty spaces.

Reviewer: The definition of “other chews” should be placed at the end of the table. The term “other chews” is included twice.

Authors: Corrected

Reviewer: The terms and definitions are still unclear to me. For example, how can the eating time be specified in the unit (n/h) and not in minutes? Please check the contents of the table again!

Authors: Corrected to – “Eating time 2 (ET2) (min/h)”

Reviewer: Line 168       The abbreviation RWS was introduce before. 

Authors: We corrected to – “The RWS..”

Reviewer: Line 168        Why is the data collection in July 2023 described in another section? (line 120)

Authors: We corrected – “The RWS was implemented from July 1 to July 31, 2023. The initial fortnight, spanning July 1 to July 14, 2023, was designated as an adjustment phase for the cows to become familiar with the RWS. The actual monitoring using RWS started on July 14, 2023, and continued until July 31, 2023.”

Reviewer: Line 180          Please explain in more detail, why the threshold of 4.70% is relevant for yourexamination.

Authors: We corrected to – “According to the literature [13] on lactose threshold, we created two groups: the first group consisted of cows with milk lactose levels below 4.70%, while the second group included cows with milk lactose levels of 4.70% or higher.

Reviewer: Line 194          Why did you use a semicolon here?

Authors: we corrected to – “chews, bolus”

Reviewer: Table 2         The table is too large. Maybe you can use landscape format?

Authors: Corrected to landscape format

 Reviewer: Could you also provide the values for milk lactose?

Authors: Group I-Milk lactose concentration < 4.70%; Group II–Milk lactose concentration ≥ 4.70%;

Reviewer:  Why are different terms used here than in the Materials and        Methods section (e.g. EC2)?

Authors: we corrected to -

Eating chews 1 (EC1)

(n/h)

Number of chews performed while the head is positioned downward during the chosen summary interval

Eating chews 2 (EC2)

(n/h)

Number of chews performed with head positioned upward during the chosen summary interval

Reviewer: Line 198          In this section, not only the significant results should be emphasized. It is equally interesting to highlight the non-significant results. Please extend this section considerably.

Authors: We added information – “Additionally, no differences were found between the groups in terms of milk fat, milk fat-to-protein ratio, eating chews 1, eating chews 2, drinking gulps, chews per minute, and activity (Table 2)”

Reviewer: Line 219          Why haven't you included a correlation table here where you can also read the non-significant values? Please also provide the other results and describe them in the text.

Authors: We added information – “Correlations between milk lactose and other investigated parameters are presented in Table 3.”Also, we added – “Table 3. Correlation between investigated parameters .

Reviewer: Line 228          You should mention in the text, that the r² is very low and could not explain very much

Authors: We corrected to –“ Correlations between milk lactose and other investigated parameters are presented in Table 3. We observed a weak significant positive correlation between milk lactose concentration and milk yield (r = 0.366, p < 0.001). Higher lactose concentrations in milk are associated with increased milk production (Figure 1). Also, we found a weak significant negative correlation between milk lactose and milk protein (r=-210, p < 0.01) (Figure 2)”

Reviewer: Line 230          General comment: There are missing some information on potential relationships between lactose content and behavior? Why should the behavior might be affected by different lactose levels? Please add some more details.

Authors: We added information in discussion section – “Coulon et al. [37] conducted research to assess the impact of walking activity on milk production and energy status in dairy farms utilizing tie-stall housing for cows. It has been observed that dairy cows exhibit behavioral signs of illness during mastitis, with alterations in activity, lying time, and feeding behavior being the primary focus of scientific study [38]. Such behavioral changes are believed to be triggered by pain or other adverse experiences [39]. In the literature, we did not find any information about the impact of milk lactose on changes in cow activity”

Reviewer: Line 230          The limitations of the study should be described in a short section (e.g. small sample size; data collection only in summer)

Authors: we added information – “This study is subject to several limitations. First, the small sample size may limit the generalizability of our findings to a broader population. Future research would benefit from including a larger and more diverse cohort of cows to validate these results. Second, data collection was confined to the summer months. Seasonal variations significantly impact the physiological conditions of dairy cows, which could in turn influence milk composition and cow behavior. Consequently, the outcomes observed in this study might not be representative of other seasons. It is recommended that subsequent studies expand the data collection period to include different seasons, ensuring a more comprehensive understanding of the dynamics explored”

Reviewer: Line 232          The first five lines provide basic information and should therefore be included in the introduction.

Authors: We moved this information into the introduction section

Reviewer: Line 245          A correlation of 0.366 should not be described as strong. Please provide a source for the classification...

Authors: We corrected to – “The correlations observed between lactose yield and milk yield align with findings by Miglior et al. [18] and Sneddon et al. [19].”

Reviewer: Line 247        Why are you citing a study here that compares fat and lactose content, while you are looking at milk yield and lactose content?

Authors: Corrected

Reviewer: Line 264          Did you investigate the influence of the lactation number in your study?

Authors: We didn't investigate this; we deleted this information

Reviewer: Line 300          How would you explain this different results in your studies?

Authors: We added information – “. Coulon et al. [37] conducted research to assess the impact of walking activity on milk production and energy status in dairy farms utilizing tie-stall housing for cows. It has been observed that dairy cows exhibit behavioral signs of illness during mastitis, with alterations in activity, lying time, and feeding behavior being the primary focus of scientific study [38]. Such behavioral changes are believed to be triggered by pain or other adverse experiences [39]. In the literature, we did not find any information about the impact of milk lactose on changes in cow activity”

Reviewer: Line 303          What is the practical impact of your findings for dairy farmers? Which benefit do they receive, if they use your method? Please add some more information

Authors: We added information – “From a practical viewpoint, we recommend regularly monitoring milk lactose levels as indicators of behavioral and physiological changes. Adjust diets for cows with lactose levels above 4.7% to maintain milk protein quality while supporting increased milk production. Implement measures to encourage physical activity in cows showing decreased movement, potentially improving overall well-being and performance”

Reviewer: Line 315          Why did you change the font size here?

Authors: Corrected

Reviewer: Line 327          There is missing the name of the journal

Reviewer: Line 329          There is missing the journal number and the pages

Reviewer: Line 331          Please use the abbreviated journal name

Reviewer: Line 337          Why do you use the term “et al.”. Please cite all authors of the publication.

Reviewer: Line 343          Is it really the correct abbreviation? “of”?

Reviewer: Line 351          Please use the abbreviated journal name

Reviewer: Line 355          Is it really the correct abbreviation? “of”?

Reviewer: Line 360          Is it really the correct abbreviation? “of”?

Reviewer: Line 374          Please use the abbreviated journal name

Reviewer: Line 379          Please delete the asterisk after the title.

Reviewer: Line 382          Is it really the correct abbreviation? “and”?

Authors: we corrected whole references list by Zotero program according journal requirements.

Reviewer 2 Report

Comments and Suggestions for Authors

General comments and suggestions

The study investigated the relationship between milk lactose concentration and feeding, rumination, and locomotion behaviour in dairy cows. The manuscript has all the sections of a standard article. However, the manuscript has drawbacks in the following sections. 

1)     The introduction needs to include key information about the study topic, for example, factors influencing milk production, feeding, and locomotion behaviour. In addition, this section needs to be streamlined, and several statements are not supported by references (check specific comments).

2)     Materials and methods need to be clearly described.

3)     Results need to be clearly presented and discussed.

Specific comments

Lines 2 – 3: The title is “in-line” necessary? Also, I would use “feeding” instead of “eating”.

Lines 13 – 23: This section can be simplified, and mentioning all the percentages here is unnecessary.

Lines 24 – 41: Streamline and should be one paragraph.

Line 38 and elsewhere in the manuscript: “Other Chews” I know that term is from the sensor, but can you be specific?

Introduction

Please streamline this section and consider adding necessary information regarding the topic, as pointed out in general comments.

See a few statements that need to be supported by references. Lines 52 – 53, 57 – 59, 65 – 67, 81 – 82, etc.

Materials and Methods

It needs to be clear and consistent throughout when the study was conducted. Check lines 120 and 168.

Also, the number of animals studied check lines 31, 131, and 177.

Lines 132 – 133: Why was it within 5 to 30 days post-calving?

Table 1: Remove duplication of other chews.

You had two groups based on lactose concentration levels: those with less than (<) 4.7% and those with greater or equal to (≥) 4.7%. Make sure you are consistent throughout the manuscript.

Results

This section also needs to be clarified. Please review Table 2 and clarify the numbers (N).

Figures 1 and 2: One decimal place should be good, and you can use full words in the Y- and X- axes instead of abbreviations.

Discussion

This section must be streamlined, and the study findings should be discussed thoroughly.

Line 286 - 296: The connection between the studied parameters and reproductive performance needs to be clearly explained.

Conclusion

Make the main conclusion more precise and supported with results.

Comments on the Quality of English Language

Check on paragraph and structure.

Author Response

Reviewer: General comments and suggestions

The study investigated the relationship between milk lactose concentration and feeding, rumination, and locomotion behaviour in dairy cows. The manuscript has all the sections of a standard article. However, the manuscript has drawbacks in the following sections. 

1)     The introduction needs to include key information about the study topic, for example, factors influencing milk production, feeding, and locomotion behaviour. In addition, this section needs to be streamlined, and several statements are not supported by references (check specific comments). 

2)     Materials and methods need to be clearly described.

3)     Results need to be clearly presented and discussed.

Authors: Authors are very thankful for the comments, which help us to improve the manuscript. All changes proposed have been included in the manuscript and highlighted in yellow and track changes.  

Best Regards, 

Prof. Ramunas Antanaitis 

Specific comments

Reviewer: Lines 2 – 3: The title is “in-line” necessary? Also, I would use “feeding” instead of “eating”.

Authors – We corrected title – “The Relation Between Milk Lactose Concentration and Rumination, Feeding and Locomotion Behavior of Fresh Dairy Cows”

Reviewer: Lines 13 – 23: This section can be simplified, and mentioning all the percentages here is unnecessary.

Authors: We corrected to – “Specifically, cows with a milk lactose concentration greater than 4.7% showed a 16.14% increase in milk yield, other chews, rumination chews, and a increase in bolus. However, these cows also experienced a decrease in milk protein concentration and a in activity levels”

Reviewer: Lines 24 – 41: Streamline and should be one paragraph.

Authors: Corrected in one paragraph.

Reviewer: Line 38 and elsewhere in the manuscript: “Other Chews” I know that term is from the sensor, but can you be specific?

Authors: We corrected information to – “These cows also exhibited changes in rumination and feeding behaviors, as recorded by the RWS: there was an increase in mastication and rumination behaviors, evidenced by a 14.09% rise in Other Chews and a 13.84% increase in Rumination Chews, along with a 16.70% boost in Bolus activity”

Introduction

Reviewer: Please streamline this section and consider adding necessary information regarding the topic, as pointed out in general comments. See a few statements that need to be supported by references. Lines 52 – 53, 57 – 59, 65 – 67, 81 – 82, etc.

Authors: We revised the entire introduction section and organized it into the following parts: 1. A brief overview of Precision Livestock Farming (PLF), including the RumiWatch system and BROLIS Sensor. 2. Identification of milk composition, with a focus on milk lactose. 3. We established a link between these two topics.

Materials and Methods

Reviewer: It needs to be clear and consistent throughout when the study was conducted. Check lines 120 and 168. 

Authors: We corrected – “The RWS was implemented from July 1 to July 31, 2023. The initial fortnight, spanning July 1 to July 14, 2023, was designated as an adjustment phase for the cows to become familiar with the RWS. The actual monitoring using RWS started on July 14, 2023, and continued until July 31, 2023.”

Reviewer: Also, the number of animals studied check lines 31, 131, and 177.

Authors: Correct number of cows – 502. We corrected in in whole manuscript.

Reviewer: Lines 132 – 133: Why was it within 5 to 30 days post-calving?

Authors: Because we focus on fresh cows. We corrected title – “The Relation Between Milk Lactose Concentration and Rumination, Feeding and Locomotion Behavior of Fresh Dairy Cows”

Reviewer: Table 1: Remove duplication of other chews.

Authors: Removed

Reviewer: You had two groups based on lactose concentration levels: those with less than (<) 4.7% and those with greater or equal to (≥) 4.7%. Make sure you are consistent throughout the manuscript.

Authors: Corrected

Results

Reviewer: This section also needs to be clarified. Please review Table 2 and clarify the numbers (N).

Authors: Corrected

Reviewer: Figures 1 and 2: One decimal place should be good, and you can use full words in the Y- and X- axes instead of abbreviations.

Authors: Corrected Figures 1 and 2.

Discussion

Reviewer: This section must be streamlined, and the study findings should be discussed thoroughly.

Line 286 - 296: The connection between the studied parameters and reproductive performance needs to be clearly explained.

Authors: We corrected discussion section and added following information –

At the beginning on discussion section – “The real-time tracking of milk lactose levels multiple times a day, allowing for the observation of its fluctuations across different physiological states and throughout the duration of cow diseases. The widespread implementation of precision farming technologies facilitates the daily documentation of individual milk profiles and variations in specific milk components. This can aid in the early detection of health issues and the initiation of prompt treatment [18].”

Description of mechanism – “Milk lactose, a significant component of bovine milk solids, is influenced considerably by the health of the udder, as well as the cow's metabolic and energy balance. Given its connections to various biological and physiological factors, the literature offers insights into milk lactose, focusing on its chemical characteristics, inheritability, and genetic links to health traits [21]. Furthermore, lactose has been identified as a marker for both subclinical and clinical ketosis [22], and it offers the most accurate assessment of energy balance [12]. Consequently, lactose concentration in milk can serve as a marker for mastitis, with notable reductions observed during inflammation [13]. Regular monitoring of lactose levels can aid in farm quality control and management, assisting in detecting low-quality milk [23] and pinpointing energy inefficiencies [14]. The measurement of lactose levels is now commonly employed as a criterion for early detection and management of herds [24].  Lactose plays a crucial role as an osmotic agent in milk, significantly influencing the transfer of water into milk from the bloodstream. Therefore, reduced lactose levels lead to a decrease in the overall volume of milk produced [12].  Glucose stimulates the growth of cells and the production of lactose in the mammary epithelial cells of dairy cows. Protein kinase B alpha functions as a metabolic regulator in the mammary gland of dairy cows, facilitating the impact of glucose on lactose production”

Explanation about lactose and activity parameters - Coulon et al. [37] conducted research to assess the impact of walking activity on milk production and energy status in dairy farms utilizing tie-stall housing for cows. It has been observed that dairy cows exhibit behavioral signs of illness during mastitis, with alterations in activity, lying time, and feeding behavior being the primary focus of scientific study [38]. Such behavioral changes are believed to be triggered by pain or other adverse experiences [39]. In the literature, we did not find any information about the impact of milk lactose on changes in cow activity”

Limitations of this study – “This study is subject to several limitations. First, the small sample size may limit the generalizability of our findings to a broader population. Future research would benefit from including a larger and more diverse cohort of cows to validate these results. Second, data collection was confined to the summer months. Seasonal variations significantly impact the physiological conditions of dairy cows, which could in turn influence milk composition and cow behavior. Consequently, the outcomes observed in this study might not be representative of other seasons. It is recommended that subsequent studies expand the data collection period to include different seasons, ensuring a more comprehensive understanding of the dynamics explored’

Conclusion

Reviewer: Make the main conclusion more precise and supported with results.

Authors: We corrected conclusion section –

“The study conclusively demonstrates a significant relationship between in-line milk lactose concentration and the rumination, eating, and locomotion behavior of dairy cows. Notably, cows with a milk lactose concentration exceeding 4.7% displayed a substantial 16.14% increase in milk yield, alongside a 14.09% increase in other chews, a 13.84% rise in rumination chews, and a 16.70% boost in bolus activity. However, these cows also experienced a 5.05% decrease in milk protein concentration and a 16.18% decline in activity change. These findings underline the importance of regular monitoring of milk lactose levels to identify behavioral and physiological changes, which could inform dietary adjustments and management practices to enhance milk production and overall cow health.

From a practical viewpoint, we recommend regularly monitoring milk lactose levels as indicators of behavioral and physiological changes. Adjust diets for cows with lactose levels above 4.7% to maintain milk protein quality while supporting increased milk production. Implement measures to encourage physical activity in cows showing decreased movement, potentially improving overall well-being and performance”

Reviewer 3 Report

Comments and Suggestions for Authors

Thank you very much for the opportunity to evaluate this rather interesting manuscript. In my opinion, the work is solid, although it requires a few corrections.

The abstract effectively presents key findings in terms of milk lactose concentrations and their correlation with cow behavior. The use of percentages and numerical values adds precision to the results.

he introduction provides a clear and concise overview of Precision Livestock Farming (PLF) and its potential benefits. The transition from PLF to the specific focus on milk attributes is smooth. The introduction highlights the significance of milk as an ideal medium for assessing the health of dairy cows. However, it could benefit from explicitly stating the existing gap in knowledge that the current study aims to address. The introduction introduces the in-line measurement of milk lactose concentration as an indicator of the health status of dairy cows. However, it would be beneficial to provide a brief context or rationale for choosing lactose concentration specifically as the focus of this study. Explicitly state the research gap or specific question that this study aims to address within the broader context of milk attributes and PLF.

The section on the registration of parameters is clear, but the subsections (2.2.1 and 2.2.2) could be better organized for improved readability. Consider restructuring the content to provide a smoother flow between the different registrations.

The results section is generally well-organized, with clear headings and subheadings. The use of tables and figures enhances the presentation of numerical data.

The discussion effectively interprets the results and integrates findings from previous studies. However, improvements in reporting precision, clarity of abbreviations, and providing more detailed explanations of biological mechanisms would enhance the overall quality of the discussion.

The conclusion is clear and succinctly summarizes the key findings of the study. It effectively communicates the main results to the reader.

Author Response

Reviewer: Thank you very much for the opportunity to evaluate this rather interesting manuscript. In my opinion, the work is solid, although it requires a few corrections.

Authors: Thank you very much for the positive comments.

Reviewer: The abstract effectively presents key findings in terms of milk lactose concentrations and their correlation with cow behavior. The use of percentages and numerical values adds precision to the results.

Authors: Thank you very much for the positive comments.

Reviewer: The introduction provides a clear and concise overview of Precision Livestock Farming (PLF) and its potential benefits. The transition from PLF to the specific focus on milk attributes is smooth. The introduction highlights the significance of milk as an ideal medium for assessing the health of dairy cows. However, it could benefit from explicitly stating the existing gap in knowledge that the current study aims to address. The introduction introduces the in-line measurement of milk lactose concentration as an indicator of the health status of dairy cows. However, it would be beneficial to provide a brief context or rationale for choosing lactose concentration specifically as the focus of this study. Explicitly state the research gap or specific question that this study aims to address within the broader context of milk attributes and PLF.

Authors: We revised the entire introduction section and organized it into the following parts: 1. A brief overview of Precision Livestock Farming (PLF), including the RumiWatch system and BROLIS Sensor. 2. Identification of milk composition, with a focus on milk lactose. 3. We established a link between these two topics.

Reviewer: The section on the registration of parameters is clear, but the subsections (2.2.1 and 2.2.2) could be better organized for improved readability. Consider restructuring the content to provide a smoother flow between the different registrations.

Authors: We included information – “During this study, we recorded milk composition using the BROLIS HerdLine in-line milk analyzer (Brolis Sensor Technology, Vilnius, Lithuania) and monitored rumination, feeding, and locomotion behavior with the help of the RumiWatch noseband sensor (RWS; ITIN + HOCH GmbH, Fütterungstechnik, Liestal, Switzerland)”

Reviewer: The results section is generally well-organized, with clear headings and subheadings. The use of tables and figures enhances the presentation of numerical data. 

Authors: Thank you very much for the positive comments.

Reviewer: The discussion effectively interprets the results and integrates findings from previous studies. However, improvements in reporting precision, clarity of abbreviations, and providing more detailed explanations of biological mechanisms would enhance the overall quality of the discussion.

Authors:

Authors: We corrected discussion section and added following information –

At the beginning on discussion section – “The real-time tracking of milk lactose levels multiple times a day, allowing for the observation of its fluctuations across different physiological states and throughout the duration of cow diseases. The widespread implementation of precision farming technologies facilitates the daily documentation of individual milk profiles and variations in specific milk components. This can aid in the early detection of health issues and the initiation of prompt treatment [18].”

Description of mechanism – “Milk lactose, a significant component of bovine milk solids, is influenced considerably by the health of the udder, as well as the cow's metabolic and energy balance. Given its connections to various biological and physiological factors, the literature offers insights into milk lactose, focusing on its chemical characteristics, inheritability, and genetic links to health traits [21]. Furthermore, lactose has been identified as a marker for both subclinical and clinical ketosis [22], and it offers the most accurate assessment of energy balance [12]. Consequently, lactose concentration in milk can serve as a marker for mastitis, with notable reductions observed during inflammation [13]. Regular monitoring of lactose levels can aid in farm quality control and management, assisting in detecting low-quality milk [23] and pinpointing energy inefficiencies [14]. The measurement of lactose levels is now commonly employed as a criterion for early detection and management of herds [24].  Lactose plays a crucial role as an osmotic agent in milk, significantly influencing the transfer of water into milk from the bloodstream. Therefore, reduced lactose levels lead to a decrease in the overall volume of milk produced [12].  Glucose stimulates the growth of cells and the production of lactose in the mammary epithelial cells of dairy cows. Protein kinase B alpha functions as a metabolic regulator in the mammary gland of dairy cows, facilitating the impact of glucose on lactose production”

Explanation about lactose and activity parameters - Coulon et al. [37] conducted research to assess the impact of walking activity on milk production and energy status in dairy farms utilizing tie-stall housing for cows. It has been observed that dairy cows exhibit behavioral signs of illness during mastitis, with alterations in activity, lying time, and feeding behavior being the primary focus of scientific study [38]. Such behavioral changes are believed to be triggered by pain or other adverse experiences [39]. In the literature, we did not find any information about the impact of milk lactose on changes in cow activity”

Limitations of this study – “This study is subject to several limitations. First, the small sample size may limit the generalizability of our findings to a broader population. Future research would benefit from including a larger and more diverse cohort of cows to validate these results. Second, data collection was confined to the summer months. Seasonal variations significantly impact the physiological conditions of dairy cows, which could in turn influence milk composition and cow behavior. Consequently, the outcomes observed in this study might not be representative of other seasons. It is recommended that subsequent studies expand the data collection period to include different seasons, ensuring a more comprehensive understanding of the dynamics explored’

Reviewer: The conclusion is clear and succinctly summarizes the key findings of the study. It effectively communicates the main results to the reader.

Authors: Thank you very much for the positive comments.

Round 2

Reviewer 1 Report

Comments and Suggestions for Authors

General comments:

The manuscript animals-2891648, entitled "The Relation Between Milk Lactose Concentration and Rumination, Feeding and Locomotion Behavior of Fresh Dairy Cows" with Mr. Antanaitis as first author was properly revised by the authors. The reviewer comments were considered in the revision and the authors left none of my questions unanswered. However, some minor inconsistencies and logical mistakes can still be found in the manuscript. Various comments for suggested changes to the text could be find in the following section (Detailed comments).

Detailed comments:

Line 8              Please delete the blanket space between „samanta.arlauskaite“ and „@lsmu.lt“

Line 15            What is the basis for this hypothesis? Please describe the possible causal relationship in simple words. It is not sufficient to state a hypothesis, but it must be substantiated (e.g. high lactose concentration, mastitis, behavioral changes).

Line 16            Please replace the term “eating” with “feeding”.

Line 24            What is the basis for this hypothesis? Please describe the possible causal relationship in simple words. It is not sufficient to state a hypothesis, but it must be substantiated (e.g. high lactose concentration, mastitis, behavioral changes).

Line 26            Please add: …investigate this relationship by using precision livestock farming

Line 35            Please harmonize within the manuscript one or two decimal numbers?

Line 45            The introduction has become much better thanks to the changes. However, the connection between lactose and behavior is still missing. Please add a few sentences on this and explain the relevance for farms (early detection by PLF).

Line 51            Please provide references for this section

Line 51            Please delete the blanket space between „impacts“ and „.“

Line 61            Please add a blanket space between “et al.” and [1].

Line 90            The abbreviation F/P ratio has not yet been introduced, so please insert it when it is first mentioned in the text.

Line 91            The amount of milk SCC should be written as a number (100,000)

Line 104          Please harmonize within the manuscript: “Other chews” or “other chews”?

Line 126          Please harmonize within the manuscript: “12,500 kg” or “12500 kg”

Line 127          Please delete the full stop after “parameters”

Line 148          Why did you refer here to Table 1? The values are not included in the table

Line 149          Why did you write the subheadings partly in capital letters?

Line 149          Please replace the term “eating” with “feeding”.

Line 151          Please harmonize within the manuscript: “locomotion” or “locomotor”?

Table 1           Please use a uniform layout in the table. The texts should all be left-aligned and have a uniform font size.

Line 166          Please report in a chronological order (first June, then July)

Table 2           Could you also provide the values for milk lactose in the table? Mean, SD, SE, Min, Max etc.? It would be interesting to see the range within the lactose levels

Why is the term “other chews” indented?

Please use the full p-value “0.001” instead of “.001

Table 3           Could you show the table on two pages? The information on the statistical text could be mentioned in the heading to save space.

Please check the hyphens and spaces in the footnotes

The contents of the table are insufficiently described in the text. Please highlight the most important findings in the text.

Figures 1/2     Abbreviations should be placed behind the words in brackets “milk yield (MY)”

Line 259          The newly inserted text is an improvement of the manuscript. However, it contains some redundancies to the previous text. Please check again critically.

Line 344   In my view, the conclusions are not logical. Why do you need to measure the lactose content in order to draw conclusions about behavioral changes, which in turn indicate physiological changes such as mastitis? With today's technology (PLF), it would be possible to draw conclusions about mastitis directly from the behavioral changes. Why does the detour via lactose make sense from your point of view? Maybe it is better to restrict your conclusion to “physiological changes”?

Line 381          Please correct: Abbreviation of the journal, Issue, pages, DOI

Line 386          Why do you use the term “et al.”? Please cite all authors of the publication.

Line 421          Please correct the name of the second author.

Line 431          Please use not the abbreviated journal name

Line 434          Please use not the abbreviated journal name

Line 448          Please use not the abbreviated journal name

Line 451          Please delete the asterisk after the title.

Author Response

General comments:

Reviewer: The manuscript animals-2891648, entitled "The Relation Between Milk Lactose Concentration and Rumination, Feeding and Locomotion Behavior of Fresh Dairy Cows" with Mr. Antanaitis as first author was properly revised by the authors. The reviewer comments were considered in the revision and the authors left none of my questions unanswered. However, some minor inconsistencies and logical mistakes can still be found in the manuscript. Various comments for suggested changes to the text could be find in the following section (Detailed comments). 

 Authors: Thank you very much for your comments and suggestions.

Detailed comments:

Reviewer: Line 8              Please delete the blanket space between „samanta.arlauskaite“ and „@lsmu.lt“

Authors: Corrected to – “[email protected]

Reviewer: Line 15            What is the basis for this hypothesis? Please describe the possible causal relationship in simple words. It is not sufficient to state a hypothesis, but it must be substantiated (e.g. high lactose concentration, mastitis, behavioral changes).

Authors: we corrected hypothesis to – “This study hypothesizes that higher in-line milk lactose concentrations are indicative of enhanced dairy cow behaviors—including increased rumination, feeding, and locomotion activities—reflecting superior overall health and well-being. It posits that fluctuations in milk lactose levels have a substantial impact on the physiological and behavioral responses of dairy cows, thereby affecting their milk yield and composition”

Reviewer: Line 16            Please replace the term “eating” with “feeding”

Authors: corrected

Reviewer: Line 24            What is the basis for this hypothesis? Please describe the possible causal relationship in simple words. It is not sufficient to state a hypothesis, but it must be substantiated (e.g. high lactose concentration, mastitis, behavioral changes).

Authors: we corrected hypothesis to – “This study hypothesizes that higher in-line milk lactose concentrations are indicative of enhanced dairy cow behaviors—including increased rumination, feeding, and locomotion activities—reflecting superior overall health and well-being. It posits that fluctuations in milk lactose levels have a substantial impact on the physiological and behavioral responses of dairy cows, thereby affecting their milk yield and composition”

Reviewer: Line 26            Please add: …investigate this relationship by using precision livestock farming

Authors: we corrected to – “The aim of this study was to investigate this relationship by using precision livestock farming”

Reviewer: Line 35            Please harmonize within the manuscript one or two decimal numbers?

Authors: Corrected to “..≥ 4.70..”

Reviewer: Line 45            The introduction has become much better thanks to the changes. However, the connection between lactose and behavior is still missing. Please add a few sentences on this and explain the relevance for farms (early detection by PLF).

Authors: we added text – “Understanding the linkage between lactose concentration and cow behavior can help farms to preemptively address issues like mastitis, which is known to alter lactose levels, before they escalate into more severe health problems. Early detection of such conditions via lactose monitoring can facilitate prompt treatment, reducing the risk of disease spread and improving milk quality. The use of PLF tools for monitoring lactose levels thus not only aids in enhancing animal welfare but also supports the sustainability of dairy operations by optimizing productivity and ensuring the production of high-quality milk. This approach exemplifies how integrating detailed behavioral and physiological data can revolutionize farm management strategies, leading to more efficient and humane dairy farming practices”

Reviewer: Line 51            Please provide references for this section

Author: We added reference for this section – “Precision livestock farming (PLF) refers to the application of real-time monitoring technologies for the management of the most granular production unit, essentially targeting individual animals through sensor technology. PLF offers significant opportunities for value creation across various stakeholders, primarily serving as an effective management resource for farmers. It enhances the capacity to boost animal welfare, efficiency, and health, while also reducing environmental impacts [1].”

Reviewer: Line 51            Please delete the blanket space between „impacts“ and „.“

Authors: Corrected

Reviewer: Line 61            Please add a blanket space between “et al.” and [1].

Authors: Corrected to – “…by Zehner et al. [2]…”

Reviewer:  Line 90            The abbreviation F/P ratio has not yet been introduced, so please insert it when it is first mentioned in the text.

Authors: We corrected to – “milk fat-to-protein ratio (F/P).”

Reviewer: Line 91            The amount of milk SCC should be written as a number (100,000)

Authors: We corrected to – “milk SCC ≥ 100,000/mL”

Reviewer: Line 104          Please harmonize within the manuscript: “Other chews” or “other chews”?

Authors: Corrected to “Other chews”

Reviewer: Line 126          Please harmonize within the manuscript: “12,500 kg” or “12500 kg”

Authors: corrected to – “12500 kg”

Reviewer: Line 127          Please delete the full stop after “parameters”

Authors: Deleted

Reviewer: Line 148          Why did you refer here to Table 1? The values are not included in the table

Authors: Deleted

Reviewer: Line 149          Why did you write the subheadings partly in capital letters?

Authors: Corrected

Reviewer: Line 149          Please replace the term “eating” with “feeding”.

Authors: Corrected to – “ Registration of Rumination, Feeding and Locomotion Behavior”

Reviewer: Line 151          Please harmonize within the manuscript: “locomotion” or “locomotor”?

Authors: We corrected to “locomotion”

Reviewer: Table 1           Please use a uniform layout in the table. The texts should all be left-aligned and have a uniform font size.

Authors: We corrected Table 1.

Reviewer: Line 166          Please report in a chronological order (first June, then July)

Authors: Corrected to – “The RWS was implemented from June 1 to July 31, 2023. The initial fortnight, spanning June 1 to July 14, 2023, was designated as an adjustment phase for the cows to become familiar with the RWS. The actual monitoring using RWS started on June 14, 2023, and continued until July 31, 2023. Data recording took place hourly, every day. Throughout the period from June 1 to June 14, 2023, during each milking session, the BROLIS HerdLine in-line milk analyzer continuously records the composition of each cow's milk from start to finish’

Reviewer: Table 2           Could you also provide the values for milk lactose in the table? Mean, SD, SE, Min, Max etc.? It would be interesting to see the range within the lactose levels

Authors: We added this information in second table.

Reviewer: Why is the term “other chews” indented?

Authors: Corrected

Reviewer: Please use the full p-value “0.001” instead of “.001

Authors: Corrected to – “<0.001”

Reviewer: Table 3           Could you show the table on two pages? The information on the statistical text could be mentioned in the heading to save space.

Authors: Corrected

Reviewer: Please check the hyphens and spaces in the footnotes

Authors: Corrected.

Reviewer: The contents of the table are insufficiently described in the text. Please highlight the most important findings in the text.

Authors: we added information – “Also, we found a weak but significant correlation between milk lactose and milk protein (r = -0.210, p < 0.01), milk fat/protein ratio (F/P) (r = -0.086, p < 0.01), eating chews 1 (r = 0.049, p < 0.05), rumination chews (r = 0.045, p < 0.05), bolus (r = 0.065, p < 0.001), chews per minute (r = 0.084, p < 0.001), and activity change (r = 0.113, p < 0.001) (Figure 2).”

Reviewer: Figures 1/2     Abbreviations should be placed behind the words in brackets “milk yield (MY)”

Authors: Corrected

Reviewer:  Line 259          The newly inserted text is an improvement of the manuscript. However, it contains some redundancies to the previous text. Please check again critically.

Authors: We corrected to – “[20]. Our findings are in agreement with the 0.40 estimate from New Zealand data presented by Sneddon et al. [20].”

Reviewer:  Line 344   In my view, the conclusions are not logical. Why do you need to measure the lactose content in order to draw conclusions about behavioral changes, which in turn indicate physiological changes such as mastitis? With today's technology (PLF), it would be possible to draw conclusions about mastitis directly from the behavioral changes. Why does the detour via lactose make sense from your point of view? Maybe it is better to restrict your conclusion to “physiological changes”?

Authors: We corrected conclusion section to – “Our study highlights significant associations between behavioral patterns and physiological changes in dairy cows, suggesting that monitoring behavioral changes could directly indicate health issues like mastitis. While our initial hypothesis explored the utility of lactose content as an indirect marker, feedback and advancements in precision livestock farming technology suggest a more direct approach might be equally or more effective. Thus, we recommend focusing on direct behavioral observations facilitated by current PLF technologies as primary indicators of physiological health. This shift acknowledges the direct link between observed behavioral changes and underlying health conditions, potentially streamlining early detection and management strategies for conditions such as mastitis’

Reviewer: Line 381          Please correct: Abbreviation of the journal, Issue, pages, DOI

Authors: Corrected to – “       Liu, P.; He, X.; Yang, X.L.; Hou, X.L.; Han, J.B.; Han, Y.H.; Nie, P.; Fang, H.; Du, X.H. Bioactivity Evaluation of Certain Hepatic Enzymes in Blood Plasma and Milk of Holstein Cows. Pak Vet J 2012., 32, 601-604, doi: http://pvj.com.pk/pdf-files/32_4/601-604.pdf”

Reviewer: Line 386          Why do you use the term “et al.”? Please cite all authors of the publication.

Authors: We corrected to – “.             Reist, M.; Erdin, D.; von Euw, D.; Tschuemperlin, K.; Leuenberger, H.; Chilliard, Y.; Hammon, H.M.; Morel, C.; Philipona, C.; Zbinden, Y.; Kuenzi, N.: J.W. Blum, J. W. Estimation of Energy Balance at the Individual and Herd Level Using Blood and Milk Traits in High-Yielding Dairy Cows1,2. Journal of Dairy Science2002, 85, 3314–3327, doi:10.3168/jds.S0022-0302(02)74420-2.’

Reviewer: Line 421          Please correct the name of the second author.

Authors: Corrected to – “Steen, A.; Østerås, O.; Grønstøl, H. Evaluation of Additional Acetone and Urea Analyses, and of the Fat-Lactose-Quotient in Cow Milk Samples in the Herd Recording System in Norway. Journal of Veterinary Medicine Series A 1996, 43, 181–191, doi:10.1111/j.1439-0442.1996.tb00443.x.”

Reviewer:  Line 431          Please use not the abbreviated journal name

Authors: We corrected to – “Lin, Y.; Sun, X.; Hou, X.; Qu, B.; Gao, X.; Li, Q. Effects of Glucose on Lactose Synthesis in Mammary Epithelial Cells from Dairy Cow. BMC veterinary research 2016, 12, 81, doi:10.1186/s12917-016-0704-x.”

Reviewer:  Line 434          Please use not the abbreviated journal name

Authors: Corrected to – “hahbazkia, H.R.; Aminlari, M.; Tavasoli, A.; Mohamadnia, A.R.; Cravador, A. Associations among Milk Production Traits and Glycosylated Haemoglobin in Dairy Cattle; Importance of Lactose Synthesis Potential. Veterinary research communications 2010, 34, 1–9, doi:10.1007/s11259-009-9324-2.

Reviewer: Line 448          Please use not the abbreviated journal name

Authors: We corrected to – “  Li, G.; Ma, X.; Deng, L.; Zhao, X.; Wei, Y.; Gao, Z.; Jia, J.; Xu, J.; Sun, C. Fresh Garlic Extract Enhances the Antimicrobial Activities of Antibiotics on Resistant Strains in Vitro. Jundishapur journal of microbiology 2015, 8, e14814, doi:10.5812/jjm.14814.”

Line 451          Please delete the asterisk after the title.

Reviewer 2 Report

Comments and Suggestions for Authors

The authors have addressed all the previous comments and suggestions, and the manuscript looks good.

Minor comments:

Line 2 – 3: In the title, the word “Fresh” does not sound good, I suggest to use “early lactation”

Please review the definition of groups based on lactose threshold lines 173 – 175. The second group is greater or equal to (≥) 4.7% and not greater than (>). Please correct this throughout the manuscript, e.g., lines 20 – 21, 347 etc.

Author Response

Reviewer: The authors have addressed all the previous comments and suggestions, and the manuscript looks good.

 Authors: Thank you very much for your comments and suggestions.

Minor comments:

Reviewer: Line 2 – 3: In the title, the word “Fresh” does not sound good, I suggest to use “early lactation”

Authors: We corrected title to – “The Relation Between Milk Lactose Concentration and Rumination, Feeding and Locomotion Behavior of Early Lactation Dairy Cows”

Reviewer: Please review the definition of groups based on lactose threshold lines 173 – 175. The second group is greater or equal to (≥) 4.7% and not greater than (>). Please correct this throughout the manuscript, e.g., lines 20 – 21, 347 etc.

Authors: Corrected